# Surface Redox Reaction for the Synthesis of NiPt Catalysts for the Upgrading of Renewable Ethanol/Methanol Mixtures

Joachim Pasel [1,*], Friederike Woltmann [1], Johannes Häusler [1] and Ralf Peters [1,2,3]

1   Institute of Energy and Climate Research, IEK-14: Electrochemical Process Engineering, Forschungszentrum Jülich GmbH, 52425 Jülich, Germany; friederike.woltmann@uni-muenster.de (F.W.); j.haeusler@fz-juelich.de (J.H.); ra.peters@fz-juelich.de (R.P.)
2   Faculty of Mechanical Engineering, Ruhr-Universität Bochum, Universitätsstr. 150, 44801 Bochum, Germany
3   Jülich-Aachen-Research-Alliance, Wilhelm-Johnen-Straße, 52425 Jülich, Germany
*   Correspondence: j.pasel@fz-juelich.de; Tel.: +49-2461-61-5140

**Abstract:** Mixtures of ethanol and methanol being synthesized from $CO_2$ and green $H_2$ can serve as sustainable base chemicals for a number of chemical processes. Amongst these processes, the catalytically supported synthesis of $CO_2$-neutral $C_4$ to $C_{10}$ alcohols is of increasing importance as, e.g., iso-butanol can be used as a drop-in fuel or after dehydration to produce iso-butene as a feedstock for the synthesis of plastics. 2-ethyl-hexanol can be further refined into solvents, tensides, or monomers. In this respect, NiPt alloys on an activated carbon support were found to be active and stable catalysts for the synthesis of iso-butanol following the Guerbet reaction scheme. In this study, two different routes are applied to the synthesis of these NiPt catalysts: a more conventional one based on the impregnation of Ni and Pt salts and an advanced path with a surface redox reaction between elemental Ni on the support and Pt ions in a polar solution. The experimental evaluation shows that the Pt particles from the surface redox reaction being exposed on the Ni particles are more active than those on the impregnated catalysts due to their high surface energy. Their specific space-time yields are 10–20 times higher.

**Keywords:** mixed alcohols; catalytic upgrading; iso-butanol; NiPt alloy catalysts; Guerbet reaction; surface redox reaction

## 1. Introduction

The human-caused increase in global temperatures had already reached 1.1 K in 2017 compared to the pre-industrial era. If emissions of the greenhouse gases responsible for global warming—above all $CO_2$—are not significantly reduced, the temperature increase by 2040 will probably total around 1.5 K [1,2]. Based on previous data and experience, it can be assumed that an increase of this magnitude would lead to major environmental damage worldwide (e.g., floods, forest fires, and severe storms, etc.) and societal tensions in regions of the world that would be particularly affected if soils dried out and water resources became scarce or dried up entirely. In response to this problem, many countries, along with the European Union, have established ambitious targets for reducing emissions of greenhouse gases in the coming decades, up to around the year 2050. In order to achieve these goals, new concepts, materials, and technologies are required in all sectors of the energy economy (heat, transport, and industry). In this context, the currently dominant approach in the transport sector is to completely dispense with carbon-containing substances as energy carriers (gasoline, diesel, and natural gas) for vehicle drivetrains and to electrify them using batteries and electric motors. Although this electrification in the mass market of passenger cars will progress in the coming years, there will still be a profound need for green and sustainable fuels wherever large masses must be transported over long distances. This applies, for example, to large cargo ships, long-haul aircraft, and long-distance road freight transport, as well as the existing car fleet, as long as the electrification

of the transport sector, which will certainly take many years, is not yet complete. In addition, there are many industrial chemical processes that cannot run without platform chemicals containing carbon. These have, thus far, relied on oil and natural gas. In order to be able to defossilize these processes, $CO_2$-neutral basic chemicals must be made available for further processing on a large scale. In this context, the focus of scientific research in the recent past has primarily been on the synthesis of the molecules of methanol [3–5], ethanol, and dimethyl ether [5–9], which are regarded as suitable basic chemicals for many chemical processes. Methanol can, e.g., serve as the starting material for the large-scale production of poly(oxymethylene) dimethyl ether fuels [10,11] or for the co-production of dimethyl carbonate, dimethoxymethane, and dimethyl ether [12]. Methanol, ethanol, and dimethyl ether can be synthesized from $CO_2$ that has been separated from the air, biogenic sources, or industrial exhaust gases and green hydrogen (from solar- or wind-powered electrolysis). Thus, they meet the requirement of $CO_2$-neutrality. In this context, the IEK-14 at the Forschungszentrum Jülich is working on the catalytically supported further processing ("value upgrading") of mixtures of green ethanol and green methanol to produce higher-quality products. The focus is on isobutanol ($C_4$), which can be used as a drop-in fuel for internal combustion engines [13] and alcohols with longer carbon chains ($C_4$–$C_{10}$). Further prominent representatives of the $C_4$–$C_{10}$ alcohols are 2-ethyl-1-hexanol, n-octanol, and n-decanol. The principle of the Guerbet reaction is used for this, in which ethanol and methanol are first dehydrogenated and then the resulting aldehydes enter into an aldol condensation with one another (see Figure 1). It was found in previous work that PtNi alloy catalysts supported on activated carbon show promising catalytic behavior for the Guerbet reaction [14]. This is why these catalysts were used for the investigations in this study.

**Figure 1.** Reaction pathway for the synthesis of iso-butanol from ethanol and methanol [15].

The specific purpose of this paper is to improve the synthesis of these PtNi catalysts and, thereby, enhance their catalytic activities. To reach this goal, the chosen scientific approach is: (i) to apply two different routes to the synthesis of NiPt alloy catalysts deposited on an activated carbon support; and (ii) to investigate their influences on the catalytic activity for the above-presented Guerbet reaction, yielding iso-butanol. In any case and independent of the synthesis route, a total weight metal loading for Ni plus Pt of 10% is aimed for. This value of 10% for the metal loading was found to be the most suitable in terms of conversion and selectivity in previous experiments on the synthesis of iso-butanol [16]. The first, more conventional synthesis route provides the impregnation of Ni and Pt salts on the surface of an activated carbon support, followed by calcination and reduction to form elemental metal particles on the surface of the support. The drawback of this route might be that the Pt particles can be hidden in the Ni bulk phase and so become inaccessible to the reactants. The second synthesis route, however, aims at generating Pt particles that are exposed on the Ni particles, thus possessing a high surface energy. It uses the so-called surface redox reaction (SRR). The SRR synthesis exploits the fact that the energy required for the exposed Pt particles to diffuse into the Ni bulk phase cannot be raised under the given reaction conditions. Thereby, it is hypothesized that the synthesis via SRR leads to

improved activity of the catalysts. The SRR synthesis starts with a catalyst, for which in the first step, only Ni is conventionally deposited on the activated carbon support via the impregnation of the Ni salt, followed again by calcination and reduction. Then, this Ni/C catalyst is given to a polar solution, in which Pt is present in ionic form. This experimental set-up now allows for the surface redox reaction between the elemental Ni and the Pt ions. This synthesis approach—also called galvanic replacement or transmetallation—was fundamentally investigated and introduced by Brankovic et al. [17,18], Sasaki at al. [19], and van Brussel et al. [20,21]. Pt has a standard potential of 1.20 V and Ni of $-0.25$ V. The difference between the two half-cell potentials is used to calculate the electrochemical potential of the overall redox reaction. In the reaction between Ni and Pt, this is 1.45 V. The difference is large enough for the redox reaction to take place voluntarily. This enables the Pt ions to be reduced to elemental Pt and the Ni to be oxidized to form $Ni^{2+}$ ions. The electrons migrate from the base Ni to the noble Pt in the polar solution, as is shown in the following reaction equation:

$$2\,Ni + [PtCl_6]^{2-} \rightarrow Pt + 2\,Ni^{2+} + 6\,Cl^- \tag{1}$$

Hereby, the $Ni^{2+}$ ions pass into the polar solution. This exchange is assumed to happen at a ratio of 1 [22]. In the literature, PtNi catalysts supported on activated carbon supports and synthesized via surface redox reactions were tested for the methanol oxidation reaction against the background of fuel cell applications [23–25]. Hu et al. [23] detected a better electrocatalytic performance with an improved Pt utilization efficiency for hollow mesoporous PtNi nanospheres, while Tamašauskaitė-Tamašiūnaitė et al. [24] showed that nano-Pt(Ni)/Ti and nano-Pt/Ti catalysts are more active with respect to the oxidation of borohydride, ethanol, and methanol compared with that of pure Pt. Wang et al. [25] found higher activity and stability for their Pt-decorated Ni nanoparticles compared to conventional Pt/C and PtRu/C catalysts. Meanwhile, Burger et al. [26] used the surface redox reaction principle to synthesize a Ni–Al catalyst with Fe doping. They observed the formation of Ni–Fe alloys, to which they attributed an increase in catalytic activity and better thermal stability.

## 2. Results and Discussion

### *2.1. NiPt Catalysts Prepared via Conventional Wet Impregnation*

In the case of the conventional synthesis route, two different samples with atomic Ni:Pt ratios of 99:1 and 95:5, respectively, were prepared. Their acronyms herein will be IMP $Ni_{99}Pt_1$/C and IMP $Ni_{95}Pt_5$/C, respectively. This section presents and discusses, in detail, the experimental results obtained with these two catalysts. At first, the findings from the catalyst characterization (X-ray Diffraction (XRD) analysis, Inductively Coupled Plasma combined with Optical Emission Spectroscopy (ICP–OES), Temperature Programmed Desorption of $H_2$ ($H_2$–TPD), and Transmission Electron Microscopy (TEM)/Scanning Electron Microscopy (SEM), combined with Energy Dispersive X-ray Spectroscopy (EDX)) are shown, followed by an explanation of the catalytic behaviors of the two samples.

### 2.1.1. XRD Analysis

Figure 2 presents the XRD patterns of the IMP $Ni_{99}Pt_1$/C and IMP $Ni_{95}Pt_5$/C catalysts, respectively, that were prepared via the above-described impregnation method. All XRD recordings from this figure show reflections between $2\Theta = 20°$ and $30°$, and at $80°$. These reflections have already been observed in previous measurements of various carbon-supported catalysts [4]. They represent the peaks of the amorphous activated carbon support itself.

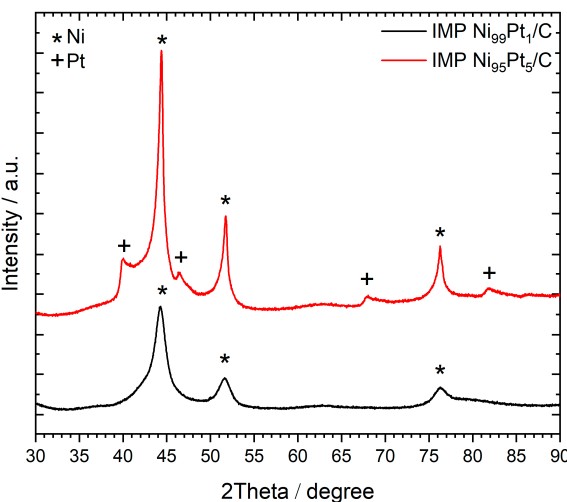

**Figure 2.** XRD patterns of the catalysts prepared via the impregnation method. Catalysts were calcined at 500 °C in $N_2$ and reduced at 250 °C in 2 vol% $H_2$ in $N_2$, with the following measuring conditions: $2\Theta = 10°$ to $90°$, and Cu K$\alpha$-radiation ($\lambda = 1.5419$ Å).

For the IMP $Ni_{99}Pt_1$/C catalyst, Ni reflections at $2\Theta = 45°$, $52°$, and $76°$ were detected, while no Pt phase can be seen in the XRD pattern. For the IMP $Ni_{95}Pt_5$/C catalyst, however, the Pt can be recognized by weak reflections at $2\Theta = 40°$ and $46°$, which are superimposed by Ni reflections at $2\Theta = 45°$ and $52°$. For both catalysts, there was no NiO-phase detectable, which would have had signals at $2\Theta = 37°$ and $43°$. These results suggest that the reduction of the catalyst in the tube furnace following impregnation was successful and that the metals were subsequently available in reduced form. It is obvious in Figure 2 that the signals of IMP $Ni_{95}Pt_5$/C are sharper than those of IMP $Ni_{99}Pt_1$/C, indicating larger and separate crystalline phases for Ni and Pt on IMP $Ni_{95}Pt_5$/C. However, the calculation of the ICP-OES-based values for at% Pt in Table 1 shows a value of only 0.4 for IMP $Ni_{99}Pt_1$/C. This is approximately one quarter of the respective Pt value for IMP $Ni_{95}Pt_5$/C in Table 1. As a result, the Pt particles on IMP $Ni_{99}Pt_1$/C could not form a separate detectable crystalline phase on the activated carbon support.

**Table 1.** Comparison of weight and atomic percentage values determined via: (i) ICP–OES measurements; and (ii) when they were calculated based on the numbers from the experimental syntheses.

| Percentage Values | IMP $Ni_{99}Pt_1$/C | | IMP $Ni_{95}Pt_5$/C | |
|---|---|---|---|---|
| | **ICP-OES** | **Calculated** | **ICP-OES** | **Calculated** |
| wt% Ni | 8.4 | 11.2 | 7.7 | 10.0 |
| wt% Pt | 0.1 | 0.3 | 0.4 | 1.5 |
| at% Ni | 99.6 | 99.2 | 98.5 | 95.7 |
| at% Pt | 0.4 | 0.8 | 1.5 | 4.3 |

### 2.1.2. ICP–OES

Table 1 depicts a comparison of the values for the weight and atomic percentages of the two impregnated NiPt catalysts when the values were determined, on the one hand, by ICP–OES measurements and, on the other, when they were calculated on the basis of the numbers from the experimental syntheses.

It becomes obvious that, in the case of the IMP $Ni_{99}Pt_1$/C catalyst, there is a significant difference between the two values for wt% Ni and wt% Pt, respectively. The intended atomic ratio of 99:1 was not exactly achieved, and it amounted to 99.6:0.4. Also, for the IMP $Ni_{95}Pt_5$/C catalyst, the intended atomic ratio could not be set and amounted to only 98.5:1.5. The observed differences in the values for the weight percentages result from the

fact that some portions of the impregnation solution were retained on the beaker glass walls and in the pipette, instead of being deposited on the activated carbon support.

### 2.1.3. $H_2$–TPD

Figure 3 presents the results of an $H_2$–TPD experiment with the IMP $Ni_{95}Pt_5/C$ catalyst. The sample was heated in a stream of Ar from 35 °C to 800 °C, where it was held for one hour. The corresponding thermal conductivity detector (TCD) signal is plotted on the left *Y*-axis, whereas the mass spectrometer signals for $H_2$ and $CO_2$ can be seen on the right *Y*-axis. Figure 3 shows a significant increase in the TCD signal normalized to the sample weight between 400 °C and 500 °C. When looking at the signals measured using a mass spectrometer, this increase was due to the desorption of substances with the $m/Z$ values of 2 for $H_2$ and 44 for $CO_2$. $CO_2$ achieved the highest signal intensity. The $H_2$ peak extends over the entire temperature range between 450 °C and 800 °C and reaches its maximum signal intensity at around 470 °C. The occurrence of the signal between 400 °C and 500 °C is explained by $H_2$ spillover from Pt to the activated carbon support at around 350 °C. In this spillover process, the $H_2$ on the Pt surface is dissociated by carboxyl and lactate groups and migrates to the surface of the activated carbon support. There, it is more strongly bound, which is why the peak only occurs at higher temperatures [27,28]. Very similar behavior during $H_2$–TPD was observed with the IMP $Ni_{99}Pt_1$ catalyst.

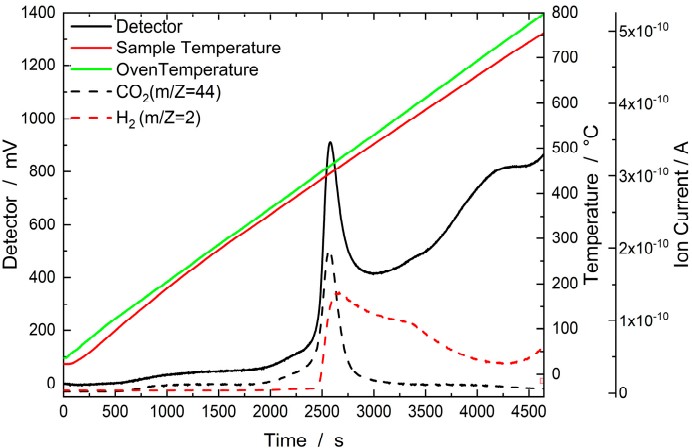

**Figure 3.** Results of the $H_2$–TPD experiment with the IMP $Ni_{95}Pt_5/C$ catalyst. The sample was heated in a stream of Ar (30 mL min$^{-1}$) from 35 °C to 800 °C, where it was held for one hour. The corresponding thermal conductivity detector signal is plotted on the left *Y*-axis, with the sample and oven temperatures on the first right *Y*-axis and mass spectrometer signals (ion currents) on the second right *Y*-axis.

### 2.1.4. TEM/SEM Combined with EDX

Figure 4 shows the TEM images of the impregnated IMP $Ni_{99}Pt_1/C$ (above) and IMP $Ni_{95}Pt_5/C$ (below) catalysts at different magnifications. In both cases, the carbon particles shown are evenly coated with metal nanoparticles. There are no local clusters. The particles are separate from each other and primarily round. The largest particles have a diameter of approximately 10 nm, whereas the value for the smallest particles is a few nanometers.

Figure 5 depicts the SEM images of the impregnated IMP $Ni_{99}Pt_1/C$ (above) and IMP $Ni_{95}Pt_5/C$ (below) catalysts at different magnifications. The largest activated carbon particles have dimensions of 80 × 60 μm. They are elongated and T-shaped with partition walls 10 μm apart. The partition walls are 2 μm wide. No pore structures are visible. The particles have a smooth surface and an irregular edge structure. The EDX analysis demonstrated an even signal intensity for both the Ni and Pt metals over the entire activated carbon particle.

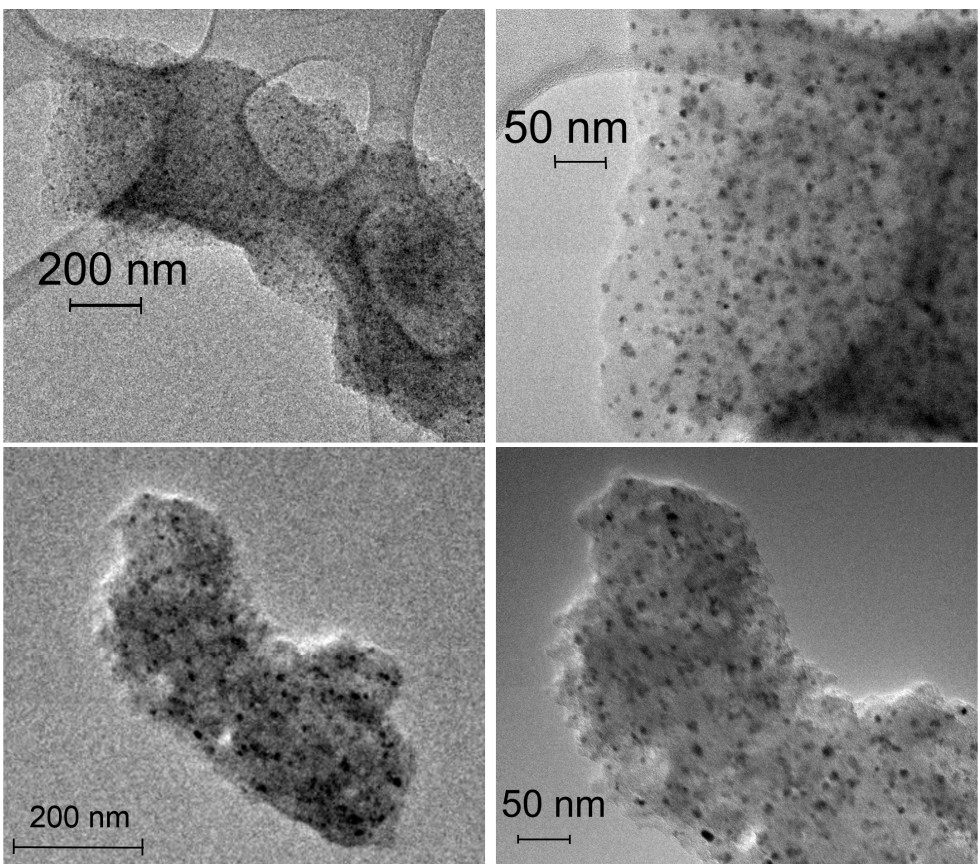

**Figure 4.** TEM images of the IMP $Ni_{99}Pt_1/C$ (**above**) and IMP $Ni_{95}Pt_5/C$ (**below**) catalysts at different magnifications.

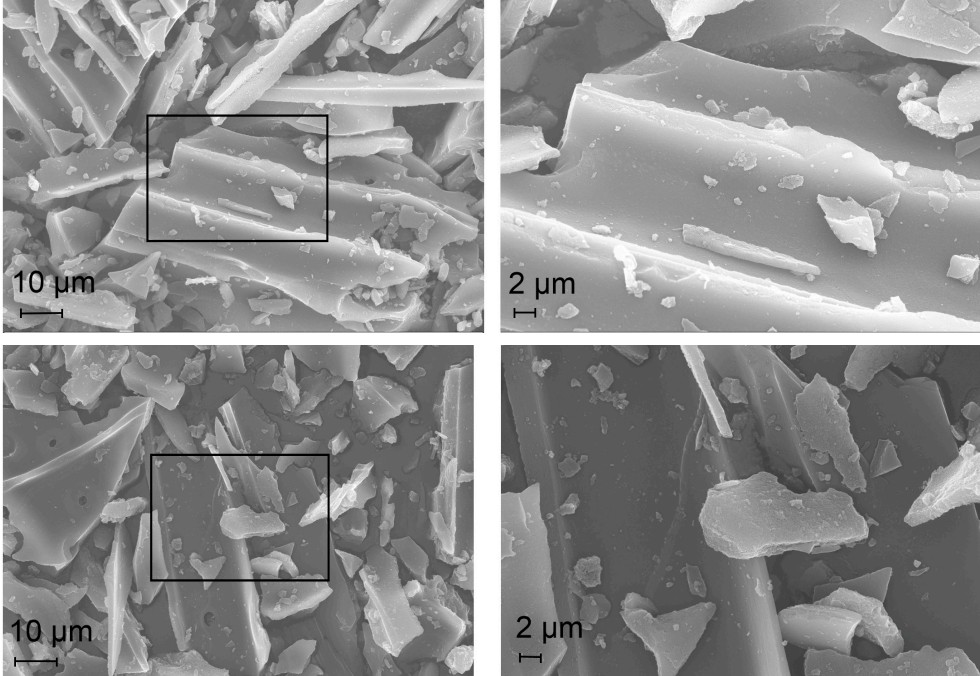

**Figure 5.** SEM images of the IMP $Ni_{99}Pt_1/C$ (**above**) and IMP $Ni_{95}Pt_5/C$ (**below**) catalysts at different magnifications.

### 2.1.5. Catalytic Activity

$$STY = \frac{n(iso - butanol)}{m(Pt + Ni) * t(reaction)} \tag{2}$$

$$m(Pt + Ni) = \frac{m(catalyst)}{wt\%(Pt + Ni)} \tag{3}$$

Table 2 presents the yields (Y) of and selectivities (S) towards the products of the reaction from Figure 1 on the two impregnated catalysts. A final iso-butanol concentration of 14.72 mmoles $L^{-1}$ was measured after four hours in the case of IMP $Ni_{99}Pt_1$/C, which corresponds to a yield of 2.45% with a selectivity of 61.03% towards iso-butanol. In addition, the specific space–time yield (STY) was calculated from the final iso-butanol concentration. For this purpose, the number of moles of generated iso-butanol molecules was related to the sum of the masses of Pt and Ni over the entire reaction period according to Equation (2). The masses of Pt and Ni were, in turn, calculated using Equation (3). Here, the weighed catalyst mass was divided by the weight percentages for Pt and Ni, which were experimentally determined via ICP–OES (cf. Table 1). For the impregnated catalyst with a Ni:Pt atomic ratio of 99:1, the specific STY was 1.02 mole $g^{-1}$ $h^{-1}$. A final iso-butanol concentration of 18.04 mmoles $L^{-1}$ was obtained with the impregnated $Ni_{95}Pt_5$/C catalyst. This corresponds to a yield of 3.01%, with a selectivity towards iso-butanol of 69.49%. The STY was 1.35 mole $g^{-1}$ $h^{-1}$. As can be seen from Table 2, the primary by-product of the Guerbet condensation was 1-hexanol with both catalysts, with selectivities towards 1-hexanol of 15.32% and 15.47%, respectively. In addition, 2-methylpropanal was synthesized with both catalysts, with a selectivity of almost 9%. 2-ethylhexane-1-ol was only synthesized on the IMP $Ni_{99}Pt_1$/C catalyst.

**Table 2.** Yields of and selectivities towards the products of the reaction from Figure 1 with the two impregnated catalysts; $m_{Cat}$ = 250 mg; reaction temperature = 165 °C; reaction time = 4 h; inlet conc. ethanol = 0.6 mole $L^{-1}$; inlet conc. NaOH = 0.45 mole $L^{-1}$.

| Reaction Products | IMP $Ni_{99}Pt_1$/C Y (%)] | IMP $Ni_{99}Pt_1$/C S (%) | IMP $Ni_{95}Pt_5$/C Y (%) | IMP $Ni_{95}Pt_5$/C S (%) |
|---|---|---|---|---|
| Iso-butanol | 2.45 | 61.03 | 3.01 | 69.49 |
| 2-methylpropanal | 0.34 | 8.55 | 0.39 | 8.94 |
| 1-propanol | 0.01 | 0.20 | 0.02 | 0.36 |
| 2-methylbutane-1-ol | 0.03 | 1.28 | 0.05 | 2.19 |
| 2-ethylbutane-1-ol | 0.02 | 1.20 | 0.03 | 2.20 |
| 1-hexanol | 0.21 | 15.32 | 0.22 | 15.47 |
| 2-ethylhexane-1-ol | 0.12 | 15.32 | 0 | 0 |

### 2.2. NiPt Catalysts Prepared via Surface Redox Reactions

To investigate the possible benefit of the surface redox reaction for the Pt doping onto the Ni-impregnated activated carbon support, a series of catalysts based on the conventionally synthesized Ni/C sample was prepared. Important criteria in optimizing the SRR catalysts were the quantity of Pt on the surface of the Ni particles supported on the activated carbon support and the amount of Ni dissolved in the polar solution. For this reason, Mintsouli et al. [22] chose a 0.1 M HCl polar solution for their investigations. As they observed, however, a large quantity of 46 wt% Ni dissolved in the 0.1 M HCl and then formed $NiCl_2$. For this work, an HCL/KCl buffer was also applied, which reduces the number of Cl ions in the solution and thus aims to minimize the Ni quantities that get lost. A third route for this study was applying a further diluted 0.01 M HCl solution with plainly fewer Cl ions and a potentially reduced formation of $NiCl_2$ in the solution.

The different catalysts are abbreviated as follows in the next sections:

- NiPt/C synthesized in an HCl/KCl buffer
    - Stirring for 15 min → SRR 15 buffer
    - Stirring for 30 min → SRR 30 buffer
    - Stirring for 60 min → SRR 60 buffer
- NiPt/C synthesized in a 0.1 M HCl solution
    - Stirring for 15 min → SRR 15 0.1 HCl
    - Stirring for 30 min → SRR 30 0.1 HCl
- NiPt/C synthesized in a 0.01 M HCl solution
    - Stirring for 5 min → SRR 5 0.01 HCl

For these catalysts, the findings from the catalyst characterization (VIS spectroscopy/ICP–OES, XRD, Thermal Gravimetric Analysis (TGA), TEM/SEM combined with EDX, and $H_2$–TPD) are shown, followed by an explanation of their catalytic behaviors.

### 2.2.1. VIS Spectroscopy/ICP–OES

The quantities of Ni in the three different buffer solutions were determined by the means of VIS spectroscopy and ICP–OES. The photometric determination was possible due to the green color of $NiCl_2$ in the buffer solution. Table 3 summarizes the obtained values for the Ni quantities in the buffer solutions after 15, 30, and 60 min of stirring. It makes clear that VIS spectroscopy is a precise, quick, and reliable method for determining Ni loss, as there is only a very small difference in the range from 1% to the values obtained via ICP–OES. It also becomes evident that the Ni values in the buffer solutions are significant and independent of the reaction time. Table S1 in the Supplementary Material shows the values for Ni and Pt weight percentages on these catalysts, when they were determined, on the one hand, via ICP–OES measurements and, on the other, when they were calculated based on the numbers derived from the experimental syntheses.

**Table 3.** Comparison of the quantities of Ni being dissolved in the polar buffer solutions measured by means of (i) Vis spectroscopy and (ii) ICP–OES.

| Catalyst | $\beta_{VIS}$ (mg $L^{-1}$) | $\beta_{ICP\text{-}OES}$ (mg $L^{-1}$) |
|---|---|---|
| SRR 15 buffer | 942 | 930 |
| SRR 30 buffer | 931 | 930 |
| SRR 60 buffer | 919 | 910 |

Table 4 shows the values for the Ni and Pt weight percentages when they were determined, on the one hand, via ICP–OES measurements and, on the other, when they were calculated based on the numbers derived from the experimental syntheses. The buffer-based SRR 15 buffer catalyst and the SRR 15 0.1 HCl and SRR 5 0.01 HCl catalysts were used. The latter two were synthesized using different HCl solutions, as proposed by Mintsouli et al. [22]. There are huge differences in the weight percentages for Ni between the data from ICP–OES and the calculated numbers in the case of the two SRR 15 buffer and SRR 15 0.1 HCl catalysts. For example, for the SRR 15 0.1 HCl catalyst, the respective values amounted to 0.10 wt% and 8.59 wt%. However, when further reducing the reaction time to 5 min and diluting the HCl solution to a value of 0.01 M, 7.33 wt% for Ni was measured via ICP–OES. This is close to the calculated value of 8.56 wt%. These reaction conditions are much more promising, as the value for Pt was also reasonable compared to the calculated one (0.24 wt% vs. 0.38).

**Table 4.** Comparison of weight percentage values for Ni and Pt on the different catalysts determined via (i) ICP–OES measurements; and (ii) when they were calculated based on the weighted portions of the educt chemicals.

| Percentage Values | SRR 15 Buffer ICP–OES | SRR 15 Buffer Calculated | SRR 15 0.1 HCl ICP–OES | SRR 15 0.1 HCl Calculated | SRR 5 0.01 HCl ICP–OES | SRR 5 0.01 HCl Calculated |
|---|---|---|---|---|---|---|
| wt% Ni | 0.32 | 8.59 | 0.10 | 8.59 | 7.33 | 8.56 |
| wt% Pt | 0.25 | 0.34 | 0.19 | 0.33 | 0.24 | 0.38 |

### 2.2.2. XRD Analysis

Figure 6 depicts the XRD patterns of the three SRR 15 buffer, SRR 15 0.1 HCL, and SRR 5 0.01 HCl samples, each of them representing the synthesis via the three different polar solutions. In the case of the SRR 5 0.01 HCl catalyst, sharp reflections at $2\Theta = 45°$, $52°$, $76°$, and $94°$ can be observed, indicating crystalline Ni phases, while, in parallel, no Pt signal was found for this sample. As Table 4 shows high values for the Ni weight loading on the surface of 7.33% in comparison to 0.24% for Pt, it is assumed that the supposable Pt reflections at $2\Theta = 40°$ and $46°$ are superimposed by the described signals for Ni. For the SRR 15 buffer and SRR 15 0.1 HCl catalysts, which both have very small quantities of Ni weight loading on their surfaces (0.32% and 0.10%, respectively), no Ni reflections were observed. However, small Pt signals were found at $2\Theta = 40°$. It can be concluded from this figure that the metal loadings on the SRR 15 buffer and SRR 15 0.1 HCl catalysts were too small to give signals at significant intensities.

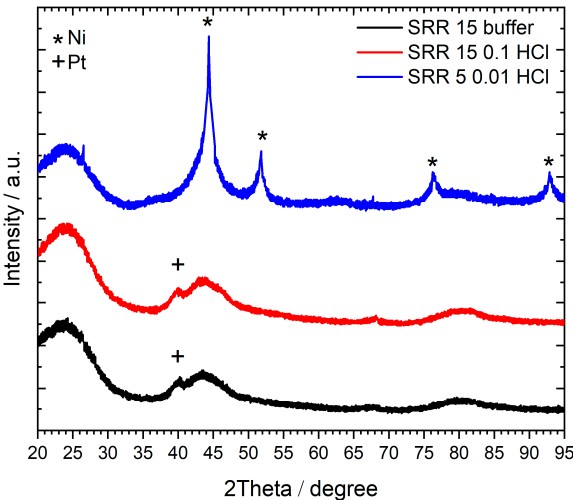

**Figure 6.** XRD patterns of the catalysts prepared via the SRR method. The measuring range was from $20°$ to $95°$ $2\Theta$ with Cu K$\alpha$-radiation ($\lambda = 1.5419$ Å).

### 2.2.3. TGA

The TGA measurements revealed, for all catalysts, a major mass loss up to a temperature of 150 °C, which can be explained by the desorption of physisorped $H_2O$. These measurements allowed for the calculation of the mass proportions of $H_2O$ of each SRR catalyst, which amounted to 5–10% in relation to the total mass of the catalyst.

### 2.2.4. TEM/SEM Combined with EDX

Figure 7 shows the TEM images of the Ni/C catalyst and of the three SRR 15 buffer, SRR 15 0.1 HCL, and SRR 5 0.01 HCl samples, each of them again representing the synthesis via the three different polar solutions. In the two upper images of the Ni/C catalyst, many small, darker points can be seen, which are stochastically distributed over the entire support. A total of 150 nanoparticles with an average diameter of 8.7 nm were identified. However, nanoparticles were not found on each of the activated carbon particles. The second line

shows the TEM images of the SRR 15 buffer catalyst. In this case, in comparison to the Ni/C catalyst, significantly fewer but also stochastically distributed nanoparticles can be seen. The average diameter of these was 7.7 nm and the total number of nanoparticles measured in the section was 15. Several stochastically distributed nanoparticles can also be seen in the TEM images of the SRR catalyst with a reaction time of 15 min in 0.1 M HCl. The total number of nanoparticles in the examined section was determined to be 25. These had an average diameter of 7.9 nm. Again, nanoparticles were not found on all the examined sections.

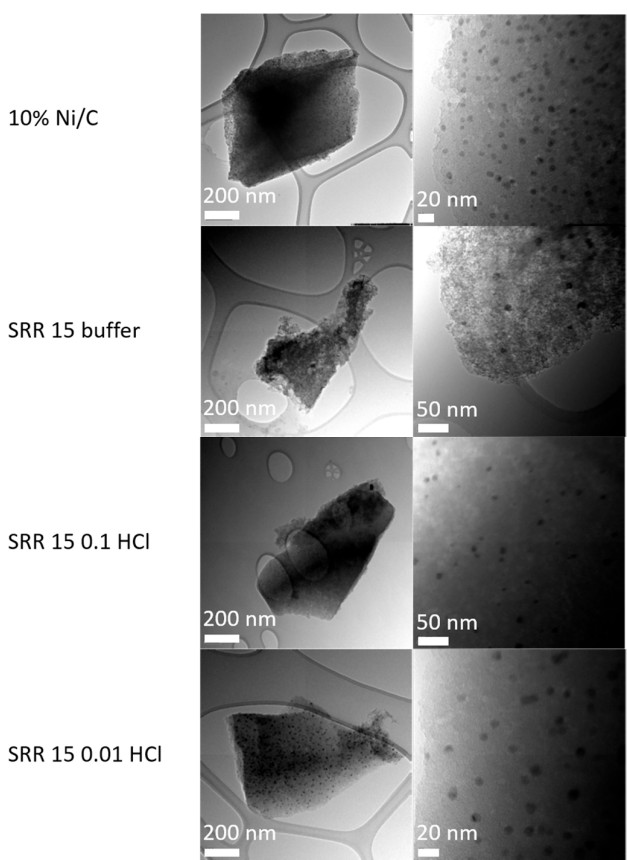

**Figure 7.** TEM images of the Ni/C catalyst and of the three samples, SRR 15 buffer, SRR 15 0.1 HCL, and SRR 5 0.01 HCl, with the scales being in the range of 20–200 nm.

Both TEM images of the SRR 5 0.01 HCl catalyst display many stochastically distributed nanoparticles. The number of nanoparticles of 150 corresponds to that of the Ni/C catalyst, whereas the other two SRR variations had significantly fewer nanoparticles. The nanoparticles had an average particle diameter of 9.5 nm. The exact size distributions are displayed in the boxplot diagram in Figure 8, which shows the maximum, minimum, and mean values determined using the TEM measurements for the four different catalysts. Overall, the nanoparticle sizes of the SRR 5 0.01 HCl catalyst were the largest, with an average diameter of 9.5 nm, whereas those of the catalyst with a stirring time of 15 min in the KCl/HCl buffer solution were the smallest at 7.7 nm.

In the SEM images in the left column of Figure 9 for the Ni/C catalyst, many elongated layer structures can be seen at lower magnifications. At the highest magnification, in the third image of this column, however, nanoparticles can be seen as small, distributed dots. As can be determined from the lowest image of this column, the Ni distribution analyzed using EDX was predominantly stochastic. A strong green coloration can be seen in the middle-left-hand part of the EDX image. In the central segment of the EDX image, Ni is shown to be slightly less green. It must be pointed out here that, in addition to the element concentrations, the color intensities in the EDX measurements also depend on the particles'

depth within the support structure. In the second column from the left, which deals with the SRR 15 buffer sample, a few clearly lighter particles that have higher atomic masses stand out. Most particles have an elongated, partially stacked shape. In addition to a few particles that are brighter overall, small bright dots can also be seen on and between the stacked structures, as is apparent in the second image from the top of this column. In this section, most of the layer structures have a length of about 100 μm, with a distance between the layers of 10–20 μm. At the highest magnification in the third image from the top, a relatively uniform surface is visible. The element mapping exhibits a non-homogeneous distribution of Ni, which accumulates on individual particles. Pt, in turn, appears to be homogeneously distributed over the entire activated carbon surface. The SEM images of the SRR catalyst with a reaction time of 15 min in 0.1 M HCl in the third column from the left again show tubular structures. Small, bright particles are spread across their surfaces. The second image from the top in this column also exhibits a very bright particle with a length of 14 μm. In the third image, which is stochastically distributed, bright particles can be seen. These particles have different sizes of up to one micrometer. The smallest have a size of less than 50 nm. A locally centered distribution of Ni cannot be seen from the EDX measurements. Both Ni and Pt appear to be stochastically distributed on the catalyst surface. In the SEM image of the SRR 5 0.01 HCl catalyst (fourth column), many layered structures can be seen, as has already been established for the other catalysts. It can be seen in the middle of the upper part of the image that these have a size of up to 200 μm in both length and width. In addition, a relatively large number of bright spots can be seen, which are mostly stochastically distributed, but sometimes accumulate. These seem to lie more on the surface of the layer structures. The second image from the top shows many similar looking, very small, bright particles and a few much larger ones, about one micron in width. These particles are less stochastically distributed and tend to accumulate in individual places. Four of the small particles can be seen in the third image. These look very similar, are round or spherical, and have diameters of 75–100 nm. As can be seen in the EDX measurement in the lowest image of the fourth column, Ni can be found on almost the entire catalyst surface. In some places, such as in the middle of the right-hand section of the image, the image exhibits a much stronger green coloration, which can be attributed to the concentration of Ni or its particle depth in the structure of the carbon support. Pt is primarily distributed stochastically on the surface. However, in some places, such as the lower left quarter of the figure, there is hardly any Ni and virtually only Pt. Overall, the SEM–EDX measurements of the various SRR catalysts show all tube/layer structures of the activated carbon support. The distribution of Pt by the SRR is stochastic for the catalysts studied. Meanwhile, the Ni loading is homogeneous for the Ni/C catalyst and the SRR 15 0.1 HCl sample. The other two catalysts do not have a homogeneous Ni distribution.

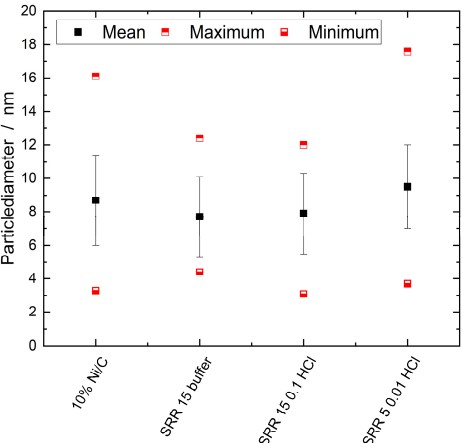

**Figure 8.** Boxplot diagram for the sizes of the nanoparticles on the Ni/C catalyst and on the three samples, SRR 15 buffer, SRR 15 0.1 HCL, and SRR 5 0.01 HCl. The error bar corresponds to the standard deviation of the mean values in positive and negative direction.

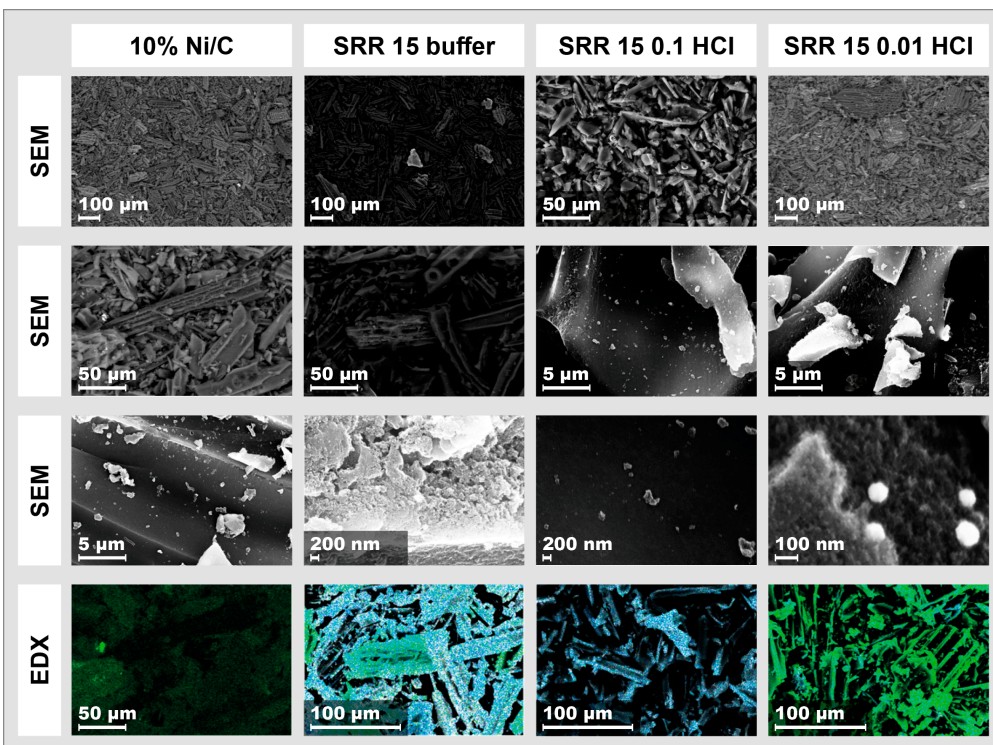

**Figure 9.** SEM and EDX images of the Ni/C catalyst and of the three samples, SRR 15 buffer, SRR 15 0.1 HCL, and SRR 5 0.01 HCl; the magnifications are between 200 and almost 250,000. In the EDX measurements, Ni is shown in green and Pt in blue.

### 2.2.5. $H_2$–TPD

All the $H_2$–TPD measurements of the SRR catalysts in Figure 10 show TCD signals at temperatures between 400 °C and 550 °C. In this range, a strong MS signal at an $m/Z$ value of 2 was detected for all the catalysts. This signal at an $m/Z$ value of 2 was due to the desorption of $H_2$. A $CO_2$ peak with an $m/Z$ value of 44 was not detected for any of the SRR catalysts.

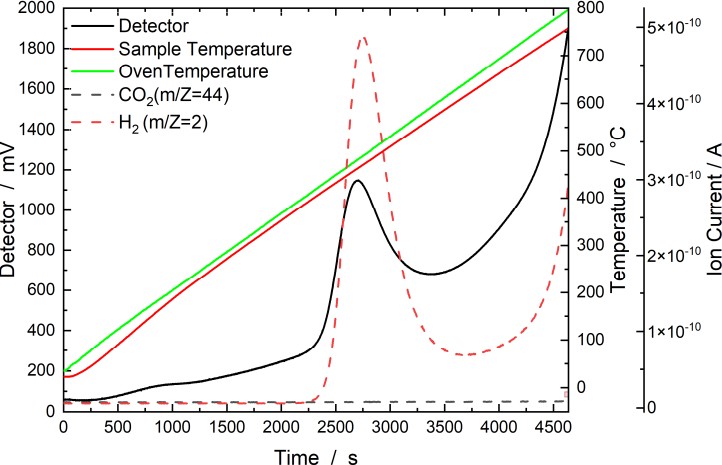

**Figure 10.** *Cont*.

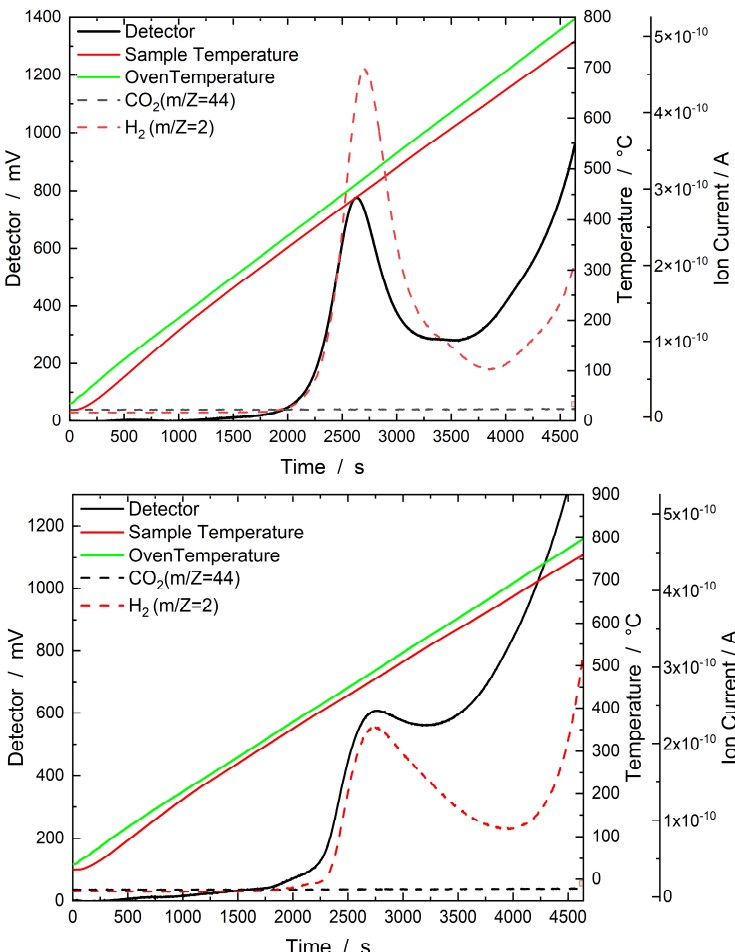

**Figure 10.** Results of the $H_2$–TPD experiment with the three samples, SRR 15 buffer (**top**), SRR 15 0.1 HCl (**middle**), and SRR 5 0.01 HCl (**bottom**). The samples were heated in a stream of Ar (30 mL min$^{-1}$) from 35 °C to 800 °C, which was then held for one hour. The corresponding thermal conductivity detector signal is plotted on the left *Y*-axis, the sample and oven temperatures on the first right *Y*-axis, and the mass spectrometer signals (ion currents) on the second right *Y*-axis.

2.2.6. Catalytic Activity

This subsection describes the catalytic behaviors of the different SRR catalysts and compares them with those of the conventionally impregnated IMP Ni$_{99}$Pt$_1$/C and IMP Ni$_{95}$Pt$_5$/C samples. The points of comparison are the specific space–time yields, the iso-butanol yields, and the selectivities towards iso-butanol. It becomes obvious that the specific STYs for the samples, IMP Ni$_{99}$Pt$_1$/C and IMP Ni$_{95}$Pt$_5$/C, amount to 1.02 and 1.35 mole g$^{-1}$ h$^{-1}$, respectively, and increase by factors of between 10 and 20 to 15.16 mole g$^{-1}$ h$^{-1}$ for the SRR 30 buffer and even 21.26 mole g$^{-1}$ h$^{-1}$ for the SRR 15 one. It can be clearly seen in the TEM images in Figures 4 and 7, respectively, that the SRR samples have much fewer Ni and Pt nanoparticles on their activated carbon supports than the two impregnated catalysts and that the particle diameters in both figures are comparable in the range of a few nanometers. Nevertheless, the SRR catalysts provide comparable iso-butanol yields of between 2% and 3%, as well as selectivities towards iso-butanol of between 60 and 70%. From these findings, the authors conclude that, for the SRR buffer catalysts, the Pt particles exposed on the Ni particles are more active than those on the impregnated catalysts due to their high surface energy. When analyzing the STYs of the catalysts, which were synthesized using the HCl solution, the value of 79.87 mole g$^{-1}$ h$^{-1}$ for SRR 15 0.1 HCl is outstanding and was partially due to the very low quantities of Ni and Pt being deposited on the activated carbon surface (cf. Table 4). The SRR 5 0.01 HCl sample,

however, represents the exact opposite in terms of the quantities of Ni, with 7.33 wt% (cf. Table 4) on its activated carbon surface. According to Equations (2) and (3), this then leads to a comparably low value for the specific STY of only 3.30 mole $g^{-1}$ $h^{-1}$. However, it can be pointed out that the number for the iso-butanol yield is highest with this sample and amounts to approximately 7% at a selectivity towards iso-butanol of 76%. This is further evidence for the higher activity of the metal particles of the SRR catalysts and demonstrates the potential of this synthesis route over the conventional one. Table 5 also shows the specific surface areas of the three samples, SRR 15 buffer, SRR 15 0.1 HCL, and SRR 5 0.01 HCl, as well as those of IMP $Ni_{99}Pt_1/C$ and IMP $Ni_{95}Pt_5/C$. They are comparable and amount to between 671 $m^2$ $g^{-1}$ and 780 $m^2$ $g^{-1}$. Thus, the specific surface areas are not decisive for the observed differences in the catalytic activities of the different samples.

**Table 5.** Specific space–time yields, iso-butanol yields, selectivities towards iso-butanol, and specific surface areas of the different catalysts; $m_{Cat}$ = 250 mg; reaction temperature = 165 °C; reaction time = 4 h; inlet conc. ethanol = 0.6 mole $L^{-1}$; and inlet conc. NaOH = 0.45 mole $L^{-1}$.

| Catalyst | Specific STY (mole $g^{-1}$ $h^{-1}$) | Y (Iso-Butanol) (%) | S to Iso-Butanol (%) | Specific Surface Areas ($m^2$ $g^{-1}$) |
|---|---|---|---|---|
| IMP $Ni_{99}Pt_1/C$ | 1.02 | 2.45 | 61.03 | 706 |
| IMP $Ni_{95}Pt_5/C$ | 1.35 | 3.01 | 69.49 | 703 |
| SRR 15 buffer | 21.26 | 2.40 | 68.68 | 780 |
| SRR 30 buffer | 15.16 | 3.01 | 67.11 | - |
| SRR 60 buffer | 17.52 | 2.04 | 70.48 | - |
| SRR 15 0.1 HCl | 79.87 | 3.49 | 74.49 | 758 |
| SRR 30 0.1 HCl | 42.07 | 6.02 | 80.09 | - |
| SRR 5 0.01 HCl | 3.30 | 6.92 | 76.19 | 671 |

As a complement to the values in Table 5, Table 6 summarizes the yields and selectivities with respect to the by-products of the Guerbet reaction for the samples, SRR 15 buffer, SRR 15 0.1 HCl, and SRR 5 0.01 HCl. The table shows that the main by-products of all the catalysts were 2-methylpropanal and 1-hexanol. These results are comparable to those obtained with the impregnated catalysts, which are displayed in Table 2.

**Table 6.** Yields of and selectivities towards the products of the reaction from Figure 1 on the catalysts, SRR 15 buffer, SRR 15 0.1 HCl, and SRR 5 0.01 HCl; $m_{Cat}$ = 250 mg; reaction temperature = 165 °C; reaction time = 4 h; inlet conc. ethanol = 0.6 mole $L^{-1}$; inlet conc. NaOH = 0.45 mole $L^{-1}$.

| Reaction Products | SRR 15 Buffer Y (%) | SRR 15 Buffer S (%) | SRR 15 0.1 HCl Y (%) | SRR 15 0.1 HCl S (%) | SRR 5 0.01 HCl Y (%) | SRR 5 0.01 HCl S (%) |
|---|---|---|---|---|---|---|
| Iso-butanol | 2.40 | 68.67 | 3.49 | 74.49 | 6.92 | 76.19 |
| 2-methylpropanal | 0.49 | 14.01 | 0.60 | 12.75 | 1.23 | 13.56 |
| 1-propanol | 0.00 | 0.04 | 0.00 | 0.05 | 0.02 | 0.23 |
| 2-methylbutane-1-ol | 0.01 | 0.83 | 0.01 | 0.39 | 0.06 | 1.35 |
| 2-ethylbutane-1-ol | 0.00 | 0.00 | 0.00 | 0.00 | 0.03 | 1.01 |
| 1-hexanol | 0.19 | 16.36 | 0.19 | 8.68 | 0.23 | 7.65 |
| 2-ethylhexane-1-ol | 0.00 | 0.00 | 0.00 | 0.00 | 0.00 | 0.00 |

## 3. Materials and Methods

### 3.1. Synthesis of the Impregnated Cataylsts

In order to produce the impregnated catalysts, the required amounts of Ni and Pt salts were first calculated for 10 wt% loading with atomic ratios of Ni:Pt of 99:1 and 95:5, respectively. To impregnate the activated carbon, the calculated amounts of $Ni(NO_3)_2 \cdot 6H_2O$ and $[Pt(NH_3)_4](NO_3)_2$ were first weighed in a test tube. The salts were then dissolved in a predefined amount of demineralized water. The solution was prepared via stirring and, if necessary, slight heating in an oil bath. After the salts were dissolved, the green

solution was added dropwise to a 100 mL Schott bottle containing the activated carbon to be impregnated. After about three drops were added, the Schott bottle was closed, and the solution was homogenized by tapping on a rubber ring. This was repeated until the complete solution was added to the activated carbon. The impregnated activated carbon was first dried in a rotary evaporator at 40 °C and a pressure of 40 mbar. The catalysts were then calcined at 500 °C in $N_2$ and reduced at 250 °C in 2 vol% $H_2$ in $N_2$. After the reduction phase, the material was cooled to room temperature overnight in the tube furnace.

### 3.2. Synthesis of the Catalysts via the Surface Redox Reaction

Three different synthesis routes were applied that differ from each other with respect to the polar solution in which the surface redox reaction took place.

### 3.2.1. KCl/HCl Buffer as a Polar Solution

The synthesis of the impregnated Ni/C base catalyst was performed analogously to the experimental procedure described in Section 3.1. The difference was that only Ni (10 wt% loading) was impregnated onto the activated carbon. Calcination and reduction were performed in the same way as described in Section 3.1. For the subsequent surface redox reaction, $K_2PtCl_6$ was dissolved in a KCl/HCl buffer solution. For this buffer, 100 mL of 0.2 M HCl was added to 400 mL of 0.2 M KCl. In the next step, the Ni/C catalyst, with a particle size smaller than 75 μm, was added to the buffer solution that also contained $K_2PtCl_6$. The suspension was first stirred at room temperature for 15, 30, and 60 min and afterwards centrifuged for 5 min at 0 °C, and then decanted. For washing, the residues were slurried three times with demineralized water, centrifuged, and then decanted. The washing solutions were stored and their absorbances were determined with a Vis spectrometer at 394 nm (see Section 3.4). The washed catalyst was then transferred to a round-bottom flask, which was placed in a water bath overnight at 65 °C. The catalyst was dried in an Ar flow within a Schlenk line, which was connected to the round-bottom flask. A small flow of Ar was passed via a cannula through an attached septum into the round-bottom flask. A second cannula was then inserted into the septum to equalize the pressure. The finished catalyst was stored under an Ar atmosphere.

### 3.2.2. 0.1 M HCl as a Polar Solution

In this variation, 0.1 M HCl was used as the reaction solution instead of the KCl/HCl buffer. The impregnated Ni/C catalyst and $K_2PtCl_6$ were added to the HCl solution. Two different batches were stirred at room temperature for 15 and 30 min, respectively. The suspensions were then centrifuged, decanted, and washed three times with demineralized water. The catalyst was then dried in a rotary evaporator at 40 °C and 30 mbar until no more water could be rotated off.

### 3.2.3. 0.01 M HCl as a Polar Solution

The only difference from the experimental procedure outlined in Section 3.2.2 was that, in this case, 0.01 M HCl was applied instead of 0.1 M HCl. In total, 5 mL of 0.2 M HCL was diluted with 95 mL of demineralized water.

### 3.3. Catalytic Experiments

To examine the catalytic activity, 250 mg of the catalysts with a grain size of $\leq$75 μm were weighed in a 100 mL autoclave. The autoclave used was a batch reactor with a reactor pot made of an Inconell 600 alloy manufactured by the Parr company. The reaction temperature, stirring speed, and switch-off pressure can be controlled by software. The actual temperature, pressure, and stirring speed were recorded by the software throughout the entire reaction. In addition to the catalyst, 70 mL of a methanolic reaction solution with 0.45 mole $L^{-1}$ of NaOH, 0.6 mole $L^{-1}$ of ethanol, and 0.015 mole $L^{-1}$ of n-decane was added to the reactor pot. The autoclave was then closed and exposed to a $N_2$ atmosphere. The reaction was carried out at 165 °C. The reaction time was four hours and started when

the target temperature of 165 °C was reached. A sample with a volume of 1 mL was taken every 30 min for the gas–chromatographic analysis. To take a sample, the shut-off valve of the capillary tube was opened. Then, 3 μL of the samples taken during the reaction was transferred into headspace vials (Thermo Scientific, Waltham, MA, USA) using a 10 μL Eppendorf pipette (Eppendorf SE, Hamburg, Germany). The vials were closed immediately afterwards. The components within the vials were separated using a gas chromatograph (Agilent 8890 GC System, Santa Clara, CA, USA) and analyzed with a mass spectrometer (GC–MS, Agilent 5977B GC7MSD). The analysis was carried out using the full evaporation method to avoid NaOH and catalyst particles getting onto the chromatography column. The chromatographic measurement took place on a 30 m long DB–Wax column with an inner diameter of 0.25 mm and a film thickness of 0.25 μm. This served as the carrier gas with a flow rate of 1.2 mL min$^{-1}$. The GC program started at a temperature of 35 °C, which held for 4 min. The mixture was then heated to 200 °C at a heating rate of 20 °C min$^{-1}$. This temperature was held again for 4 min. Calibration solutions made from the pure substances were used to quantify the reaction products. N-decane was used as an internal standard.

### 3.4. Vis Spectroscopy

To determine the $NiCl_2$ content in the reaction solutions, the respective extinctions were measured with a VWR Visible Spectrophotometer PV4 (VWR International, Radnor, PA, USA) at a wavelength of 394 nm. For quantification, six different calibration solutions with mass concentrations of $NiCl_2$ from 1400 mg L$^{-1}$ to 100 mg L$^{-1}$ were prepared and measured.

### 3.5. TGA

A PerkinElmer STA 6000 (Perkin Elmer, Waltham, Ma, USA) was used to determine the water content of the catalysts. Approximately 10 mg of the sample was weighed into an $Al_2O_3$ crucible and heated in two stages. First, the sample was heated from 30 °C to 150 °C at a heating rate of 15 K min$^{-1}$. This temperature was held for 15 min. The sample was then heated from 150 °C to 500 °C again with a heating rate of 15 K min$^{-1}$ and held at 500 °C for 15 min. It was then cooled to 100 °C at a rate of 40 K min$^{-1}$. During the measurement, the $N_2$ flow was 20 mL min$^{-1}$. To determine the background, an empty $Al_2O_3$ crucible was measured using the same temperature program.

### 3.6. H$_2$–TPD

H$_2$–TPD was performed in three steps. First, during the rinsing phase, Ar flowed over the catalyst surface at a temperature of 35 °C with a rate of 30 mL min$^{-1}$. H$_2$ was then adsorbed on the catalyst surface in single pulses. Twenty pulses of H$_2$ with a fixed volume were pulsed at a constant temperature of 35 °C. There was a time span of 2 min between the pulses. In the third step, H$_2$–TPD took place. The sample was heated from 35 °C to 800 °C under an Ar flow of 30 mL min$^{-1}$ at a heating rate of 100 K min$^{-1}$. The temperature was maintained at 800 °C for one hour. TPDRO 1100 Series apparatus from Thermo Fischer Scientific (Waltham, MA, USA) was used.

### 3.7. ICP–OES

For the ICP–OES measurements, 4–30 mg of the catalyst samples was leached with 4 mL of aqua regia at 80 °C for two hours and then diluted to a total volume of 50 mL. Each of the solutions was diluted between 10- and 100-fold. In addition, the reaction solutions from the surface redox reactions were analyzed by diluting each sample 1000-fold, 100-fold, and 10-fold. The samples were measured using a Thermo Scientific iCAP 7600.

### 3.8. XRD

A Bruker D8 DISCOVER (Billerica, MA, USA) with a LYNXEYE XE–T detector was used for the XRD measurements. The measurements were carried out with Cu–Kα radiation (λ = 0.15419 nm). The distance between the sample and radiation source was 430 mm. The

measurements were carried out with a step size of 0.08° at 0.5 steps per second in the range of 10° ≤ 90°. The voltage was 40 kV and the current 40 A. A 2.5° Soller slit was used in the primary beam path.

*3.9. $N_2$ Sorption*

The measurements concerning the specific surface areas of the different catalysts were performed according to the description by Pasel et al. [5]. The Brunnauer–Emmett–Teller methodology was used [29].

*3.10. TEM*

The TEM measurements for Figure 7 were carried out using a Philips CM 200 field emission gun transmission electron microscope (Philips, Amsterdam, Netherlands). Thereby, the catalysts were suspended in iso-propanol, a drop of which was placed on a 3 mm Cu grid. The grid was coated with an ultra-thin carbon film and air-dried overnight at room temperature before the analysis was undertaken. The acceleration voltage was 200 kV. The TEM images from Figure 4 were recorded using an FEI Technai G2 F20 transmission electron microscope (FEI Company. Hillsboro, OR, USA) with an acceleration voltage of 200 kV. The samples were dissolved in an iso-propanol/water mixture and then homogenized in an ultra-sonic bath for 15 min. Two droplets of the solution were then deposited on a Cu grid. The grid was dried overnight in an oven at 60 °C.

*3.11. SEM/EDX*

A Zeiss Gemini Ultra Plus (Zeiss, Jena, Germany) was used to measure the signal from the scanning electron microscope. An acceleration voltage of 20 kV and a working distance of 8.5 mm were utilized in all measurements.

## 4. Conclusions

The technique of surface redox reactions in a polar solution is described in the literature as a promising route for preparing catalysts with exposed metal particles possessing high surface energies. Thereby, the electrochemical potential of the overall redox reaction must be large enough for the noble metal to be reduced and the less noble one to be oxidized. In this context, this work concentrated on the concerted deposition of Pt particles on the surface of Ni particles supported on activated carbon via the surface redox reaction. The experimentally determined specific space–time yield was defined as the measure of these Pt particles' catalytic activity. It was observed in this work that the specific space–time yield of the NiPt catalysts for the iso-butanol synthesis from methanol and ethanol mixtures increased by a factor of between 10 and 20 compared to the conventional impregnation synthesis for NiPt/C. The experiments also make it clear that, during the catalyst synthesis, undesired side reactions like the formation of $NiCl_2$ from Ni and Cl ions must be suppressed by choosing the appropriate synthesis conditions. Otherwise, active catalyst species will finally be lost in the polar solution.

**Supplementary Materials:** The following supporting information can be downloaded at: https://www.mdpi.com/article/10.3390/catal14010077/s1, Table S1. Comparison of the weight percentage values for Ni and Pt determined via: (i) the ICP–OES measurements; and (ii) when they were calculated based on the weighed portions of the educt chemicals.

**Author Contributions:** Conceptualization, J.H.; funding acquisition, R.P.; investigation, F.W.; methodology, J.H.; supervision, J.P. and R.P.; writing—original draft, J.P.; writing—review and editing, J.P., J.H. and R.P. All authors have read and agreed to the published version of the manuscript.

**Funding:** This study was funded by the Deutsche Forschungsgemeinschaft (DFG, German Research Foundation)–491111487.

**Data Availability Statement:** Data are contained within the article and Supplementary Materials.

**Acknowledgments:** Special thanks are due to the fuel synthesis team at Jülich and all project and cooperation partners. The authors would also like to thank Birgit Schumacher and Denise Günther for recording the SEM and TEM images.

**Conflicts of Interest:** The authors have no conflicts of interest to disclose. The funders had no role in the design of the study; in the collection, analyses, or interpretation of data; in the writing of the manuscript; or in the decision to publish the results.

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
