# Peer review of "Surface Redox Reaction for the Synthesis of NiPt Catalysts for the Upgrading of Renewable Ethanol/Methanol Mixtures"

_catalysts, doi:10.3390/catal14010077_

Round 1

Reviewer 1 Report

Comments and Suggestions for Authors

The article “Surface redox reaction for the synthesis of NiPt catalysts for the upgrading of renewable ethanol/methanol mixtures” by Joachim Pasel et al is devoted to the synthesis of new catalysts of the Guerbet reaction and their approbation. The research as a whole is interesting, and a number of successful reactions have been carried out. However, the article contains a large number of significant shortcomings and errors. The reviewer's main comments are summarized below.

Abstract. Error in line 21. Their specific space-ime yields were 10–20 times higher.

The Materials and Methods section should be moved before the Results and Discussion section.

What is the reason for loading metals at 10% by weight?

The Introduction should be modified to highlight the novelty of your research compared to already used catalysts. The text should be more logical in content.

What does Pt(Ni) stand for in equation (1)? With this form of notation, the balance of the equation is not maintained.

Figure 1 (Scheme) is very low resolution. In Fig. 4 (SEM) the scale is completely invisible.

The text of the manuscript contains incomprehensible phrases “Error! Reference source not found."

The header text is 3.2.2.0.1 . M HCl as a polar solution error

The English language of the manuscript should be improved. The text contains a number of grammatical errors and inaccuracies.

p. 3.4 UV-Vis spectroscopy?

In Fig. 2 the phases described should be designated. Why is there a difference in the diffraction patterns of the IMP Ni99Pt1/C and IMP Ni95Pt5/C samples, although their phase composition is the same and the % ratio is quite close (judging by ICP)?

According to tables 2 and 3, the method used to obtain catalysts is poorly reproducible and ineffective. The difference in platinum content compared to the calculated one is almost 300 times. This is due to the use of a buffered acid solution. In Fig. 1 phase designations should be applied. Diffraction patterns in their present form are uninformative.

The order of the imagines is very mixed up. The article is more like a draft.

Comments on the Quality of English Language

The English language of the manuscript should be improved. The text contains a number of grammatical errors and inaccuracies.

Reviewer 2 Report

Comments and Suggestions for Authors

The authors reported a new method focused on the concerted deposition of Pt particles over the surface of Ni particles supported on activated carbon via the surface redox reaction. The paper at this stage demands a major revision, and after that it might be taken into consideration for acceptance.

Comment 1: Authors are highly recommended for careful proofreading before the formal submission. For example, the typo “Error! Reference source not found” appeared in the paper for 5 times, which should be related with specific Figure or Table in the main text. In Table 1, data in rows 3 and 4 should refer to “at% Pt” and “at% Ni”, respectively.

Comment 2: For Table 1, as authors mentioned, significant difference between “ICP-OES experimental value” and “calculated value”, which means the data itself is questionable, and it failed to prove the data reliability. On the following section, authors used buffer solution plus dilute HCl solution (Table 3 and Table 4) to narrow the observed difference between “ICP-OES experimental value” and “calculated value”, this information and its related process is recommended to be transferred into the Supporting Information part, because the process how to improve the data quality can easily distract audience’s attention. Leaving one catalyst prepared by conventional impreganation method and one premium catalyst developed via surface redox reaction in the main text is sufficient, which helps the audience to focus on the main topic of this paper (SRR is much better than conventional method).

Comment 3: As shown from Figure 3, the selected 4 catalysts exhibited minimal difference in the mean of particle diameter and error bar (please provide the specific definition of the selected error bar). Then the question is, why the catalyst “SRR 15 buffer” exhibited significantly enhanced space-time yield for iso-butanol formation? Is chloride anion the key factor for enhanced space-time yield over Ni and Pt particles? Does Ni-Pt alloy form during the SRR preparation?

Round 2

Reviewer 1 Report

Comments and Suggestions for Authors

After re-reading the article, the reviewer had the following questions and comments:

1. What does the reflex in the area 2θ=27º in Fig. 6 for sample SRR5 0.01HCl correspond to?

2. In Fig. 9, the scale bar is completely indeterminable for all SEM-EDX images.

3. Authors should add references to the Introduction. For example, the statements on lines 51-53 clearly require references. The given list of references is generally outdated. The newest work is dated 2020. A link to the authors' previous studies cited is required (line 73). Very negligent attitude towards references to literature.

4. The purpose of the work in the Introduction should be highlighted and emphasized.

Reviewer 2 Report

Comments and Suggestions for Authors

The authors have properly addressed my comments, and now the revised manuscript exhibits improved logical flow and convincing derivation. Therefore, the revised manuscript can be accepted in the present form.

Author Response

Thanks for accepting our paper in its revised version.